# Efficacy of 3 Tesla Functional Magnetic Stimulation for the Treatment of Female Urinary Incontinence

**DOI:** 10.3390/jcm11102805

**Published:** 2022-05-16

**Authors:** Andrea Braga, Fabiana Castronovo, Giorgio Caccia, Andrea Papadia, Luca Regusci, Marco Torella, Stefano Salvatore, Chiara Scancarello, Fabio Ghezzi, Maurizio Serati

**Affiliations:** 1Department of Obstetrics and Gynecology, EOC—Beata Vergine Hospital, 6850 Mendrisio, Switzerland; fabiana.castronovo@eoc.ch (F.C.); giorgio.caccia@eoc.ch (G.C.); 2Faculty of Biomedical Sciences, Università della Svizzera Italiana, 6900 Lugano, Switzerland; andrea.papadia@eoc.ch; 3Department of Obstetrics and Gynecology, EOC—Civico Hospital, 6900 Lugano, Switzerland; 4Department of General Surgery, EOC—Beata Vergine Hospital, 6850 Mendrisio, Switzerland; luca.regusci@eoc.ch; 5Department of Gyanecology, Obstetric and Reproductive Science, Second University of Naples, 80100 Naples, Italy; marcotorella@iol.it; 6Department of Obstetrics and Gynecology, IRCSS San Raffaele Scientific Institute, 20132 Milan, Italy; stefanosalvatore@hotmail.com; 7Department of Obstetrics and Gynecology, Del Ponte Hospital, University of Insubria, 21100 Varese, Italy; chiarascanca@gmail.com (C.S.); fabio.ghezzi@uninsubria.it (F.G.); mauserati@hotmail.com (M.S.)

**Keywords:** functional magnetic stimulation, extracorporeal magnetic innervation, chair, female urinary incontinence, stress urinary incontinence, overactive bladder

## Abstract

Functional magnetic stimulation (FMS) is a new technique for the conservative treatment of Urinary incontinence (UI), based on magnetic induction. It induces controlled depolarization of the nerves, resulting in pelvic muscle contraction and sacral S2-S4 roots neuromodulation. The aim of this study was to assess the efficacy of the new 3 Tesla FMS chair, both in patients with pure stress urinary incontinence (SUI) and in women with pure overactive bladder (OAB) symptoms. A prospective observational study was conducted in our urogynaecologic unit. All the patients involved were consecutive women with pure SUI or pure OAB symptoms treated by a 3 Tesla electromagnetic chair. The primary outcome was a subjective outcome evaluation by the PGI-I Scale and a patient-satisfaction scale. The secondary outcome was the change score of the UDI-6, IIQ-7, ICIQ-SF and OAB-q SF questionnaires from baseline to final visit. At 2 months follow-up, 28 out of 60 patients (47%) with SUI symptoms and 20 out of 40 patients (50%) with OAB symptoms declared themselves cured. Considering cured and improved patients, the subjective cure rates were 68.3% (41/60) and 70% (28/40) for patients with SUI and OAB symptoms, respectively. The results of this study showed that the 3 Tesla electromagnetic chair may be an effective option for the treatment of UI.

## 1. Introduction

Urinary incontinence (UI) is a common health problem with a negative impact on female quality of life (QoL). The prevalence of UI in the female population is variable. In the literature, it has been reported to be as high as 55%, and the trend increases with aging [1,2,3]. The guidelines of the leading urological and urogynaecological societies recommend pelvic floor muscle training (PFMT) as the first-line treatment for different types of UI [4]. In the last years, new technologies and applications, such as electrical stimulation, radiofrequency, laser therapy and pulsed magnetic stimulation, have been introduced in this field as an alternative or in addition to classical rehabilitation.

FMS is a technique based on Faraday’s law of magnetic induction, approved by the United States Food and Drug Administration (FDA) in 1998, for stimulating the central and peripheral nervous system [5,6]. It generates electrical activity, which induces controlled depolarization of the nerves, resulting in pelvic muscle contraction and sacral S2-S4 roots neuromodulation [5]. For this peculiarity, it has been applied for the treatment of all types of UI. In addition, the setting (patient sitting on a chair with clothes) and the lack of direct activation of skin sensory receptors and C-fibres, which can cause pain and discomfort, make this procedure more comfortable than conventional electrical stimulation [5,7]. However, few studies in the literature have assessed the efficacy of this device on female UI treatment, with different protocols and different outcome measures [8]. Furthermore, the devices used developed dissimilar magnetic field power up to a maximum of 2.5 Tesla. Similar to PFM and Electric Functional Stimulation, the FMS chair could be an effective and safe procedure in all types of incontinence, with high patient acceptance. Patients were seated comfortably in the chair and fully clothed. This feature is an advantage, especially for the elderly population.

The aim of this study is to assess the efficacy of the new 3 Tesla electromagnetic chair stimulation in patients with stress urinary incontinence (SUI) and overactive bladder (OAB) symptoms.

## 2. Materials and Methods

This is a prospective study performed in our urogynaecological unit, namely the EOC—Beata Vergine Hospital, Mendrisio, Switzerland, between January 2020 and September 2021. We enrolled all consecutive women who complained of pure SUI and OAB symptoms according to the International Urogynecological Association (IUGA)/International Continence Society (ICS) terminology for Female Pelvic Floor Dysfunctions [9], for at least 3 months. Exclusion criteria were as follows: pregnancy, implanted pacemaker or cardioverter defibrillator, implants made of ferromagnetic metal at or near the site of stimulation, clinically significant voiding dysfunction, postvoid residual volume >100 mL, previous pelvic surgery for the treatment of pelvic organ prolapse (POP) or of UI, neurological diseases, mixed urinary incontinence (MUI) symptoms, pelvic organ prolapse quantification (POPQ) ≥ stage II, documented recurrent urinary tract infections, and previous pharmacological treatment for OAB during the last 3 months.

At the initial visit, patients underwent medical history collection, physical examination, urine laboratory analyses and post-void bladder ultrasound. A stress test was performed in the lithotomy and upright positions with a full bladder (ultrasonographic measurement > 300 mL). All women also completed the following questionnaires before and after treatment: Urogenital Distress Inventory Short Form (UDI-6), Incontinence Impact Questionnaire Short Form (IIQ-7), International Consultation on Incontinence Questionnaire Short Form (ICIQ-SF), and OAB-questionnaire Short Form (OAB-q SF).

All patients deemed eligible for the treatment were scheduled for 3 Tesla electromagnetic chair (FMS Tesla Care ^®^, Iskra Medical d.o.o., Ljubljana, Slovenia).

It is based on Faraday’s law of magnetic induction, whereby a time-varying magnetic field induces electrical activity that depolarizes the nerves and causes the pelvic floor muscles to contract or relax. Repeated activation of the terminal motor nerve fibres and the motor end plates tend to build muscle strength and endurance [8,10]. The main stimulation targets are the afferent branches of the pudendal nerve to inhibit the detrusor muscle through central reflexes and the efferent nerve branches to facilitate strengthening of the pelvic floor muscles and increase the tonus of the urethral sphincters, thereby inhibiting the detrusor muscle through the guarding reflex [10].

The FMS treatment was administered for 20 min per session, twice per week for a total of 8 weeks. We used specific therapy program for different types of UI as suggested by producers and based on previous experiences [8] (Table 1). During the treatment session, the electric impulse intensity was adjustable according to the patient’s tolerance.

Before and during treatment, physiotherapists with specialized training in pelvic floor dysfunctions followed these patients. The primary outcome was subjective outcomes evaluation. All women completed the Patients Global Impression of Improvement (PGI-I) Scale [11], and a patient-satisfaction scale (a single, self-answered, Likert-type scale of 0–10 that grades the patient’s degree of satisfaction regarding continence: 0 represents “not satisfied”, and 10, “satisfied”) [12]. Subjective success was indicated both by “very much improved or much improved” (PGI-I ≤ 2) and by a patient-satisfaction score ≥ 8, while subjective improvement was indicated both by “minimally improved” (PGI-I ≤ 3) and by a patient-satisfaction score ≥ 7. The secondary outcome was the change score of the UDI-6, IIQ-7, ICIQ-SF and OAB-q SF questionnaires from baseline to final visit. The Declaration of Helsinki was followed, and pretreatment written informed consent for the FMS procedure was obtained from all the patients in this observational prospective evaluation. The study does not require ethical/institutional review board approval because normal clinical practices have been followed [13].

## 3. Statistical Analysis

Statistical analysis was performed with IBM-SPSS v.17 for Windows (IBM Corp, Armonk, NY, USA). Descriptive statistics were used to describe basic patients’ characteristics. Non-parametric paired samples test was used to compare results before and after FMS treatment. Pearson’s correlation was used to perform the correlation analysis. A *p* value < 0.05 was used to define statistical significance. The Cox proportional hazards model was used for univariate analysis to evaluate factors potentially affecting the risk of failure during the study period. Statistical significance was considered achieved when *p* < 0.05.

## 4. Results

One hundred consecutive women (60 with pure SUI and 40 with OAB symptoms), who fulfilled the inclusion criteria, had undergone FMS treatment, were considered. Baseline patients characteristics are summarized in Table 2. No statistically significant differences were found between the study groups.

At 2 months follow-up (16 treatments), all patients were available for evaluation. No patients were lost to follow-up. No adverse effects have been reported.

Twenty-eight out of sixty patients (47%) with SUI symptoms and 20 out of 40 patients (50%) with OAB symptoms declared themselves cured. Considering cured and improved patients, the subjective cure rates were 68.3% (41/60) and 70% (28/40) for patients with SUI and OAB symptoms, respectively. No statistically significant differences were found between the study groups (Table 3).

Furthermore, statistically significant differences were found between patients’ reported outcomes pre- and post-procedure. Post-treatment questionnaires scores were lower than pre-treatment scores for both SUI and OAB symptoms (Table 4).

Figure 1 and Figure 2 displays pre- and post-treatment changes for SUI and OAB symptoms.

Table 5 and Table 6 reports univariable analysis of factor potentially involved in the risk of FMS failure for SUI and OAB symptoms respectively. We did not find any risk factor statistically associated with the FMS failure.

## 5. Discussion

The present study, to the best of our knowledge, evaluated for the first time in the literature, the subjective outcomes of the 3 Tesla functional magnetic chair device in women with UI. We found that FMS might be an effective and safe procedure in patients who complain of both SUI and OAB symptoms. Subjective cure rates were found in 47% and 50% of patients with SUI and OAB, respectively. In addition, when we consider the rate of improvement, the effectiveness is higher, 68.3% for patients with SUI and 70% for patients with OAB symptoms.

Although the US FDA has approved extracorporeal FMS for UI treatment since 1998, few studies in the literature have assessed the efficacy of safety for this procedure. Lukanovi’c et al. [8], in a recent systematic review on the effectiveness of magnetic stimulation in the treatment of UI, which included articles published between 2010 and 2020, showed that only 12 studies were eligible. These studies, which have mainly considered patients with SUI, used different devices with various magnetic field power (up to a maximum of 2.5 Tesla), different diagnostic methods to define the type and severity of UI and different tools to evaluate outcomes (standardized questionnaires are sometimes used).

In addition, the authors reported the results of their clinical prospective non-randomized experience for 82 patients with UI, treated with FMS and assessed by a standardized ICIQ-UI SF questionnaire. They found an improvement in terms of UI and ICIQ-UI SF score after treatment, regardless of UI type, especially in women with SUI.

Changes in UI by ICIQ-SF and changes in the number of pads used per day, were also used by Samuels et al. [14] to evaluate 75 women with all types of UI. The average improvement of 49.93% in ICIQ-SF score was observed after the sixth treatment, which further increased to 64.42% at follow-up. The highest level of improvement was reached in patients suffering from MUI (69.90%). Furthermore, a highly significant medium correlation (r = 0.53, *p* < 0.001) was found between the ICIQ-SF score improvement and the reduction in pad usage.

Weber-Rajek et al. [15], in a randomized controlled trial, compared 40 women who underwent 12 sessions of pelvic floor muscle training (PFMT) with 37 women who received 12 sessions of extracorporeal magnetic innervation for SUI symptoms. In both groups, a statistically significant decline in depressive symptoms and an improvement in UI and quality of life were found.

Another comparative study of three different treatment methods for SUI:FMS, Electromyographic (EMG) biofeedback and PFMT by Özengin et al. [16] showed that all methods were effective of increasing pelvic flooor muscle (PFM) strength. This reduced UI symptoms and improved Qol.

Doğanay et al. [17], in a prospective study, evaluated long-term effects of extracorporeal magnetic innervations (ExMI) on 68 women with SUI and 69 women with urge incontinence. At 6 months after 16 sessions of ExMI, 32 (47%) patients with SUI were totally dry in a negative stress test, and 27 (39%) showed improvement in the frequency of daily leak episodes from 3.2 times to 1.2 times. In the urge incontinence group, 40 (58%) patients were dry, and 18 (26%) significantly improved the average number of incontinence episodes decreased from 3.7 times to 1.7 times per day. However, beneficial effects are temporary, and there is high recurrence rate (53% at 6 months).

Our study seems to show that the 3 Tesla FMS chair can be associated with excellent subjective satisfaction not only in SUI patients, but also in those with OAB symptoms. The higher cure and improvement rate found in these patients could be related to higher magnetic power than previous devices analysed in literature. Moreover, our electromagnetic chair has two magnetic field generators placed on the seat and on the back, which are powered and controlled by an external power supply. The second (back) generator makes it easier for sacral S–S4 roots neuromodulation, which could explain the positive effect on OAB symptoms.

Further points of strength include: (1) a highly homogeneous study population with the exclusion of women with mixed incontinence, (2) a clinical evaluation performed in all patients, (3) the subjective outcomes evaluation using four standardized questionnaires, and (4) no patients were lost to follow-up.

We acknowledge that the weaknesses of this study could be (1) the limited sample size, (2) the lack of objective evaluation, and (3) the lack of randomization and/or control group. However, we also emphasize that no studies in the literature have assessed 3 Tesla devices thus far, no larger investigations evaluated both SUI and OAB symptoms, and few studies used standardized questionnaires.

## 6. Conclusions

The results of this study showed that the 3 Tesla electromagnetic chair (FMS Tesla Care, Iskra Medical^®^) may be an effective option for the treatment of SUI and OAB symptoms, with great patient acceptance and no side effects.

The treatment is painless, and the patients were seated comfortably on the chair fully clothed. This characteristic represents an advantage, especially for elderly population.

## Figures and Tables

**Figure 1 jcm-11-02805-f001:**
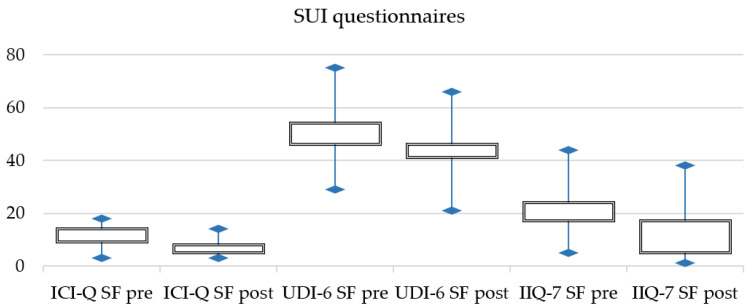
SUI changes in patients reported outcomes.

**Figure 2 jcm-11-02805-f002:**
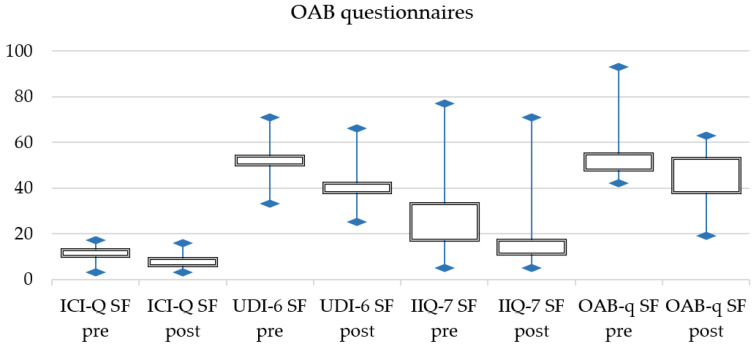
OAB changes in patients reported outcomes.

**Table 1 jcm-11-02805-t001:** FMS Tesla Care^®^ chair program for different type of UI.

Programs	Frequency (Hz)	Time (s)	Pulsed Time (μs)	Active Time (s)	Passive Time (s)	Therapy Time (min)
SUI	35	12	300	6	6	30
OAB	10	12	250	6	6	30

Hz: Hertz; s: seconds; min: minutes.

**Table 2 jcm-11-02805-t002:** Baseline patients characteristics.

Patients Characteristics	SUI (60)	OAB (40)	*p* Value
Age, yr, median, (IQR)	52 (52–63)	63 (63–72)	0.06
BMI, kg/m^2^, median, (IQR)	26 (26–29)	29 (29–30)	0.23
Menopausal, no. (%)	24 (48)	25 (83.3)	0.05
HRT, no. (%)	4 (8)	4 (13.3)	0.71
Previous vaginal deliveries, median, (IQR)	1 (1–3)	2 (1–3)	0.88
Macrosome, ≥4000 g, no. (%)	2 (4)	5 (16.7)	0.11
Operative delivery, vacuum/forceps, no. (%)	3 (6)	5 (16.7)	0.22
Cesarean delivery, no. (%)	4 (8)	2 (6.7)	1.00
Recurrent Urinary Tract Infection, no. (%)	4 (8)	2 (6.7)	1.00

IQR: Interquartile Range; BMI: Body Mass Index; HRT: Hormonal Raplacement Therapy.

**Table 3 jcm-11-02805-t003:** Cure and Improvement rate at 2 months follow-up.

Patients Symptoms	Cure Rate % (*n*/*n*)	Cure and Improvement Rate % (*n*/*n*)
SUI	47 (28/60)	68.3 (41/60)
OAB	50 (20/40)	70 (28/40)
	*p* value 0.84	*p* value 1.00

*n*: number.

**Table 4 jcm-11-02805-t004:** Changes in patients reported outcomes at 2 months follow-up.

Questionnaire	SUI Pre (m/IQR)	SUI Post (m/IQR)	*p* Value **	OAB Pre (m/IQR)	OAB Post (m/IQR)	*p* Value **
ICI-Q SF	9 (9–14)	5 (5–8)	0.001	11 (10–13)	6 (6–9)	0.001
UDI-6 SF	46 (46–54)	41 (41–46)	0.001	50 (50–54)	38 (38–42)	0.001
IIQ-7 SF	17 (17–24)	6 (5–17)	0.001	17 (17–33)	11 (11–17)	0.001
OAB-q SF	-	-		48 (28–55)	38 (38–53)	0.001

m: median; IQR: interquartile range. ** Pearson’s chi-squared test.

**Table 5 jcm-11-02805-t005:** Univariable analyses of variables potentially involved in the risk of FMS failure for SUI.

Characteristics	Cured (*n* = 28)	Not Cured (*n* = 32)	*p* Value **
Age, year, median (IQR)	49 (33–63)	55 (45.5–67)	0.21
Menopause, *n* (%)	15 (53.5)	14 (43.7)	0.60
BMI, kg/m^2^, median (IQR)	24.7 (21.8–30.1)	24.6 (24.28–27.5)	0.72
UDI-6 SF pre, median (IQR)	45.8 (41–56)	47.9 (43.7–54)	0.89
ICIQ-SF pre, median (IQR)	10 (7–14)	12.5 (7–14.5)	0.71
IIQ-7 SF pre, median (IQR)	16.5 (0–30)	22 (11–38.2)	0.25

IQR: interquartile range. ** Univariate Cox proportional hazards model.

**Table 6 jcm-11-02805-t006:** Univariable analyses of variables potentially involved in the risk of FMS failure for OAB.

Characteristics	Cured (*n* = 20)	Not Cured (*n* = 20)	*p* Value **
Age, year, median (IQR)	56 (53–73)	67.5 (51.5–73)	0.66
Menopause, *n* (%)	5 (25)	3 (15)	0.69
BMI, kg/m^2^, median (IQR)	24.8 (24–29)	29.3 (23.8–31.35)	0.86
UDI-6 SF pre, median (IQR)	50 (44–62)	45.8 (41.6–60)	0.45
ICIQ-SF pre, median (IQR)	10 (8–15.5)	10 (3.5–15)	0.36
IIQ-7 SF pre, median (IQR)	16.5 (5.5–49.5)	16.5 (14–58)	0.79
OAB-q SF pre, median (IQR)	52 (48–67)	46 (37.5–74)	0.78

IQR: interquartile range. ** Univariate Cox proportional hazards model

## Data Availability

Data available on request due to privacy restrictions.

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
