# Peer review of "Efficacy of 3 Tesla Functional Magnetic Stimulation for the Treatment of Female Urinary Incontinence"

_jcm, 2022, doi:10.3390/jcm11102805_

Round 1
Reviewer 1 Report
The current study is a prospective observational study, that evaluates the effect of 3 Tesla functional magnetic stimulation for the 2 treatment of female urinary incontinence.
The authors should consider the following comments on their manuscript.
Please describe more about the rationale of study.
The method section for functional magnetic stimulation is unclear. Please provide more detail about it.
In the result section, please provide the time point to measure the outcomes, and add them to the tables, too.
The used statistical method for data analysis should be added to the legends of tables.
It is better to provide a comparison between two different populations.
In tables 5 and 6, the percentage for menopause women is missed.
An RCT is recommended for future studies and should be considered as the limitations of the study.
Please provide ethical considerations for the study.
Author Response
We would like to thank the reviewer for their thoughtful and supportive comments. We have revised our manuscript accordingly:
Reviewer 1.
The current study is a prospective observational study, that evaluates the effect of 3 Tesla functional magnetic stimulation for the 2 treatment of female urinary incontinence. The authors should consider the following comments on their manuscript.
Please describe more about the rationale of study.
We added in the text as follows:
“Similar to PFM and Electric Functional Stimulation, the FMS chair could be an effective and safe procedure in all types of incontinence, with high patient acceptance. Indeed, patients were seated comfortably in the chair and fully clothed. This feature is an advantage, especially for the elderly population.”
The method section for functional magnetic stimulation is unclear. Please provide more detail about it.
We added in methods as follows: It is based on Faraday's law of magnetic induction, whereby a time-varying magnetic field induces electrical activity that depolarizes the nerves and causes the pelvic floor muscles contraction or relaxation. Repeated activation of the terminal motor nerve fibers and the motor end plates will tend to build muscle strength and endurance [8; 10]. The main stimulation targets are the afferent branches of the pudendal nerve to inhibit the detrusor muscle through central reflexes, and the efferent nerve branches to facilitate strengthening of the pelvic floor muscles and increase the tonus of the urethral sphincters, thereby inhibiting the detrusor muscle through the guarding reflex.
In the result section, please provide the time point to measure the outcomes, and add them to the tables, too.
We had already indicated in the result section the time point to measure the outcomes (2 months follow-up – 16 treatments). We reported it in the legend of table 3 and 4 as suggested.
The used statistical method for data analysis should be added to the legends of tables.
We reported the statistical methods into the legend of tables.
It is better to provide a comparison between two different populations
We compared the two different populations without finding significant statistical differences (table 2 -3).
In tables 5 and 6, the percentage for menopause women is missed.
We apologises for the missing data. We reported the percentage in the tables
An RCT is recommended for future studies and should be considered as the limitations of the study.
We added in the discussion as limitation the lack of randomization and/or control group
Please provide ethical considerations for the study.
We reported in the methods that our study does not require Ethical/Institutional review board approval because normal clinical practice has been followed.
- Rosmini F. Ferrigno L.(2015). Ethical aspects of epidemiological research. Istisan reports 15/44. Epidemiology and Public Health. National Institute of Health 2384-8936 (Italy).

Reviewer 2 Report
It is a valuable report. The authors state that there have been no previous reports of 3 Tesla FMS chairs.
The authors also describe some limitations, and although the paper has some problems, I think the content is not bad.
Journal of Clinical Medicine is an open-access journal, and I think it is acceptable for the content to be adopted if the authors can resolve the ethical issues.
Author Response
We would like to thank the reviewer for their thoughtful and supportive comments. We have revised our manuscript accordingly:
Reviewer 2.
It is a valuable report. The authors state that there have been no previous reports of 3 Tesla FMS chairs.
The authors also describe some limitations, and although the paper has some problems, I think the content is not bad. Journal of Clinical Medicine is an open-access journal, and I think it is acceptable for the content to be adopted if the authors can resolve the ethical issues.
We reported in the methods that our study does not require Ethical/Institutional review board approval because normal clinical practice has been followed.
- Rosmini F. Ferrigno L.(2015). Ethical aspects of epidemiological research. Istisan reports 15/44. Epidemiology and Public Health. National Institute of Health 2384-8936 (Italy).
Round 2
Reviewer 1 Report
My comments responded satisfactorily.
Reviewer 2 Report
In my country, even retrospective studies require the approval of a research ethics board. However, since I am not an expert in the international background of ethics committees, I will leave it to the editorial board members to decide whether or not to accept the article for publication.